# Transarterial Chemoembolization Outperforms Radioembolization in Early- and Intermediate-Stage Hepatocellular Carcinoma: A Multicenter Retrospective Study

**DOI:** 10.3390/cancers17132254

**Published:** 2025-07-07

**Authors:** Faisal M. Sanai, Adnan Alzanbagi, Mohammed Arabi, Sarah S. Alfawaz, Khalid I. Bzeizi, Mohammed Almatrafi, Abdulmalik M. Alsabban, Jameel Bardesi, Hamdan S. Alghamdi, Mohamed Shawkat, Talal M. Alotaibi, Khairat H. Alameer, Shadi Saleem, Saad Abualganam, Abdulaziz M. Tashkandi, Noha H. Guzaiz, Nesreen H. Abourokbah, Hassan O. Alfakieh, Majed Almaghrabi, Abeer A. Alabdullah, Lujain H. Aljohani, Nuwayyir A. Alqasimi, Saad Aldosari, Azzam Khankan, Dieter Broering, Saleh A. Alqahtani

**Affiliations:** 1Gastroenterology Section, Department of Medicine, King Abdulaziz Medical City, King Abdullah International Medical Research Center, Jeddah 21423, Saudi Arabia; 2Liver Disease Research Center, College of Medicine, Riyadh 11461, Saudi Arabia; 3Gastroenterology Department, King Abdullah Medical City, Makkah 24246, Saudi Arabia; 4Interventional Radiology, King Abdulaziz Medical City, Ministry of National Guard Health Affairs, Riyadh 22490, Saudi Arabia; 5Organ Transplant Centre of Excellence, King Faisal Specialist Hospital and Research Center, Riyadh 12713, Saudi Arabia; kbzeizi@kfshrc.edu.sa (K.I.B.); dbroering@kfshrc.edu.sa (D.B.);; 6Department of Medical Imaging, King Abdulaziz Medical City, Jeddah 21423, Saudi Arabia; 7College of Medicine, University of Jeddah, Jeddah 23218, Saudi Arabia; 8Department of Hepatobiliary Sciences and Organ Transplantation, Hepatology Section, Ministry of National Guard Health Affairs, Riyadh 22490, Saudi Arabia; 9Department of Medicine, Gastroenterology Section, King Faisal Specialist Hospital and Research Center, Riyadh 12713, Saudi Arabia; 10Radiology Department, King Faisal Specialist Hospital and Research Center, Riyadh 12713, Saudi Arabiasabualganam@kfshrc.edu.sa (S.A.); 11Department of Medical Imaging and Intervention, King Abdullah Medical City, Makkah 24246, Saudi Arabia; guzaiz.n@kamc.med.sa; 12College of Medicine, King Saud Bin Abdulaziz University for Health Sciences, Jeddah 21423, Saudi Arabia; 13Liver, Digestive, and Lifestyle Health Research Section, King Faisal Specialist Hospital and Research Center, Riyadh 12713, Saudi Arabia; aaalabdullah@kfshrc.edu.sa; 14College of Medicine, Princess Nourah Bint Abdulrahman University, Riyadh 11564, Saudi Arabia; 15Division of Gastroenterology and Hepatology, Weill Cornell Medicine, New York, NY 10065, USA

**Keywords:** hepatocellular carcinoma, BCLC, transarterial chemoembolization, transarterial radioembolization, Y90, locoregional therapy, survival, response, treatment, disease control, decompensation

## Abstract

This study compares two treatments for patients with unresectable liver cancer: Transarterial Radioembolization (TARE), a treatment that delivers radioactive microspheres directly to a liver tumor, targeting it with radiation, and Transarterial Chemoembolization (TACE), a procedure that delivers chemotherapy directly to a liver tumor while blocking its blood supply to enhance the treatment's effectiveness. The goal was to understand which treatment provides better survival outcomes. A total of 279 patients were included in the analysis, with 104 receiving TARE and 175 receiving TACE. A key finding was that average survival was better for patients who received TACE compared to TARE. Specifically, in early-stage liver cancer, TACE led to a longer survival (60 vs. 25 months). TACE also had higher response rates in early-stage patients, with more achieving a complete response (43.2% vs. 34.3%). Patients treated with TARE were more likely to experience liver complications, like hepatic decompensation, than those treated with TACE. The study concluded that TACE was found to provide better survival outcomes, especially for patients with early-stage liver cancer, and had fewer complications. This suggests that doctors should carefully consider which embolization therapy to use based on the stage and condition of the patient.

## 1. Introduction

Hepatocellular carcinoma (HCC) is the third leading cause of cancer-related mortality in men and the sixth most common malignancy worldwide [1]. Due to comorbidities from cirrhosis and late diagnosis, only 30–40% of patients are eligible for curative treatments like liver transplantation (LT) or surgical resection [1,2]. Locoregional therapies (LRTs) are critical for patients with unresectable HCC (uHCC), particularly in early (Barcelona Clinic Liver Cancer [BCLC] 0/A) and intermediate (BCLC B) stages. In BCLC B, transarterial chemoembolization (TACE) is the standard of care [2,3].

However, TACE efficacy is limited by the heterogeneity of intermediate-stage HCC [3,4]. Transarterial radioembolization (TARE) is another LRT involving the infusion of Yttrium-90 (Y90) radioisotope into tumor-feeding arteries, enabling high-dose localized radiation while sparing the surrounding liver tissue. TARE is recognized as an alternative for managing BCLC A and B disease, especially in those with non-tumoral portal vein thrombosis [2,5]. In addition, the recent BCLC 2022 staging update proposes TARE as a curative option for early-stage (BCLC 0/A) HCC when ablation or resection is infeasible [3].

In clinical practice, not all patients with HCC are eligible for the optimal treatment recommended for their disease stage due to technical issues or coexisting comorbidities. In this context, the roles of the two principal LRTs, TACE and TARE, have been explored across different stages of HCC, with conflicting reports of comparative efficacy. Therefore, this multicenter retrospective study aims to compare the effectiveness and safety of TACE versus TARE in early- and intermediate-stage HCC.

## 2. Patients and Methods

### 2.1. Study Design and Participants

This multicenter, retrospective, observational study evaluated the efficacy and safety of TACE and TARE in patients aged ≥18 years with BCLC 0, A, or B staged HCC. From January 2016 to March 2023, consecutive, unselected patients with uHCC undergoing embolization therapies were included from four tertiary centers in Saudi Arabia: King Abdulaziz Medical City—Jeddah, King Faisal Specialist Hospital and Research Center—Riyadh, King Abdallah Medical City—Makkah, and King Abdulaziz Medical City—Riyadh. Data were collected from the Saudi Observatory Liver Disease (SOLID) registry or from clinical records.

Patients were eligible if they had a diagnosis of HCC, either radiologically or histologically confirmed [2]; were classified as BCLC stage 0, A, or B; had an Eastern Cooperative Oncology Group (ECOG) performance scores of 0–2; had adequate liver function (Child–Pugh A or up to B9); were ineligible for surgical resection or percutaneous ablation; had at least one tumor lesion accurately measured by computed tomography (CT) or magnetic resonance imaging (MRI) per modified Response Evaluation Criteria in Solid Tumors (mRECIST); and were treated with either TACE or TARE. Exclusions included other simultaneous local treatments, such as ablation, surgery, radiotherapy, transarterial bland embolization, or hepatic arterial infusion chemotherapy (HAIC); incomplete clinical data; BCLC C or D; previous or concurrent malignancies distinct from HCC; advanced renal failure requiring dialysis; high-grade portosystemic encephalopathy (PSE); those who had undergone prior LT; or those who were infected with human immunodeficiency virus (HIV).

Treatment decisions were based on tumor phenotype, including tumor size, location, and number. The decision to treat with either TACE or TARE was consensus-based during weekly multidisciplinary HCC meetings represented by medical oncology, hepatology, transplant or hepatobiliary surgery, and interventional radiology.

### 2.2. Transarterial Therapy Procedures

Conventional TACE (c-TACE) involved super-selective cannulation of the tumor-feeding artery and infusion of an emulsion of 5–30 mg doxorubicin (Actavis, Dublin, Ireland) and 0.5–15.0 mL lipiodol (Laboratoire Guerbet, Paris, France). TACE with drug-eluting beads (DEB-TACE) used a mixture of 100–300-micron beads loaded with 75 mg doxorubicin. Embolization aimed to stagnate antegrade flow. Additional Gelfoam or particles were used as needed. Treated lesions were based on background liver function. Accessory supply from extrahepatic arteries was carefully evaluated to minimize the risk of residual or recurrent disease. In this study, TACE was performed using either c-TACE or DEB-TACE techniques according to operator preference and institutional protocols. Additional TACE sessions were based on responses evaluated via follow-up imaging and clinical status.

TARE delivered radioactive Y90 microspheres, which were injected via the hepatic artery into target tumor-feeding vessels. TARE was performed after an angiogram with technetium-99m macroaggregated albumin (MAA) to assess hepatic perfusion, exclude extrahepatic uptake, calculate lung shunt fraction, and determine the appropriate dose. In our study, two types of radioactive intra-arterial microspheres were used: glass spheres (TheraSphere; Boston Scientific, Marlborough, MA, USA) and resin spheres (SIR-Spheres; SIRTex, Woburn, MA, USA). Dosing used the body surface area (BSA) method, the Medical Internal Radiation Dose (MIRD) model, or the partition model. Treatments were super-selective (radiation segmentectomy) with curative intent or non-selective lobar infusion for palliative intent. Bilobar treatments used sequential lobar sessions 4–6 weeks apart when liver function was stable.

Retreatment decisions were based on multiphase CT or MRI at 8–12 weeks for TACE and 12–24 weeks for TARE in the first year, then 12–24 weeks thereafter. Complete treatment was defined as the treatment of all tumors identified on baseline imaging. In patients with multifocal bilobar disease, either the right or left liver was treated initially (staged treatments), embolizing the remaining liver at a second session to complete treatment.

### 2.3. Follow-Up and Safety Analyses

Demographic and clinical data were collected and updated at each center. The primary endpoint was overall survival (OS), defined as the time from initial transarterial treatment to death or the last follow-up. Secondary endpoints included tumor response assessed by mRECIST criteria by radiologists blinded to the clinical data and survival outcomes. Imaging (dynamic CT or MRI) occurred at 8–12 weeks in the first year, then 12–24 weeks thereafter. Treatments were repeated up to three times for TACE and twice for TARE, or until unacceptable toxicity, disease progression, or futility, as ascertained by the treating physician. Adverse events (AEs) were graded per Common Terminology Criteria for Adverse Events (CTCAE, version 5.0) at every contact with the patient based on medical records, laboratory findings, or imaging. Treatment-related AEs (TRAEs) were physician-assessed. Safety outcomes, including all-cause and TRAEs, as well as hospitalization, were evaluated for up to 6 months post-procedure.

### 2.4. Statistical Analysis

Demographics were summarized with descriptive statistics, using counts and percentages for nominal data, and means ± standard deviation (SD) or medians with interquartile range (IQR) for continuous variables. Nominal data used Fisher’s exact test or chi-square test, and continuous variables used independent *t*-test or Mann–Whitney U test, as appropriate. Overall survival (OS) in the TACE and TARE groups was analyzed with Kaplan–Meier curves and compared with the log-rank tests. Hazard ratios (HRs) for survival outcomes and their 95% confidence intervals (CIs) were calculated. Analyses used SPSS v28.0. A two-sided *p* < 0.05 was deemed statistically significant.

### 2.5. Ethical Considerations

The study was conducted in accordance with the guidelines of the Declaration of Helsinki and the principles of Good Clinical Practice, and the institutional review boards of the participating centers approved the study. As this was a retrospective study using de-identified data, the requirement for informed consent was waived in accordance with local regulations.

## 3. Results

### 3.1. Baseline Characteristics and Interventions

From 2016 to 2023, 357 patients with unresectable HCC were screened, with 279 enrolled, including those undergoing TACE (n = 175) or TARE (n = 104) (Appendix A). Demographics and characteristics are in Table 1. The overall cohort (mean age 67.6 ± 10.1 years, 207 [74.2%] male) included 108 (38.7%) patients with BCLC 0/A disease and 171 (61.3%) with BCLC B. Viral etiology (62.7%), diabetes (59.5%), and Child–Pugh A (75.3%) were common. ECOG 2 status (12.5%), ascites (14.0%), variceal bleeding (0.72%), and PSE (0.72%) were noted.

Treatment cohorts were balanced (Table 1). Median α-fetoprotein (AFP) levels were similar in both treatment cohorts, although more patients in the TARE cohort had AFP ≥400 ng/mL (20.4% vs. 11.0%, *p* = 0.032). Additionally, patients in the TARE cohort had more lesions (median 2.0 vs. 1.0, *p* < 0.001) than those undergoing TACE, and a larger maximum tumor diameter (4.4 [2.9–6.4] vs. 3.1 [2.0–4.2] cm, *p* < 0.001). Cohorts were comparable in disease etiology, cirrhosis, and performance status. Rates of ascites (13.7% vs. 14.4%), variceal bleeding (1.1% vs. 0%), and PSE (0.6% vs. 1.0%) were similar. TARE patients were more likely to have BCLC B (69.2% vs. 56.6%, *p* = 0.036), with diabetes (66.3% vs. 55.4%) trending higher (*p* = 0.072). In BCLC 0/A, TARE-treated patients had higher median AFP levels (8.9 vs. 8.0, *p* = 0.046) and a larger maximum tumor diameter (4.2 [2.3–5.8] vs. 2.7 [1.8–3.7] cm, *p* = 0.001). Other characteristics were similar (Table 2). In BCLC B patients, diabetes was more common with TARE (63.9%) than TACE (47.5%, *p* = 0.033), and patients had larger tumors (maximum diameter 5.0 [2.7–7.8] vs. 4.0 [3.0–5.6] cm, *p* = 0.049) (Table 3).

Prior to the index procedure, 45 (16.1%) patients received additional interventions, including ablation in 40 (14.3%) patients (TACE, 28 [16.0%]; TARE, 12 [11.5%]) and segmental resection in 5 (1.8%) patients (TACE, 3 [1.7%]; TARE, 2 [1.9%]). After the index TACE procedure, 63 (36%) of the patients underwent additional embolization therapies. Of these, 31 (49.2%) had a repeat TACE, 16 (25.4%) had TARE as a subsequent modality, and 16 (25.4%) received both TACE and TARE. In the TARE cohort, 35 (33.7%) had additional embolization therapies, including TACE as a subsequent modality in 8 (22.9%), repeat TARE in 20 (57.1%), and 7 (20%) with both TACE and TARE. LT was performed in 24 patients overall, with nearly identical rates between the two groups: 15 (8.6%) of patients in the TACE cohort and 9 (8.7%) in the TARE cohort.

### 3.2. Treatment Outcomes and Survival

By May 2024, the median follow-up duration of the overall cohort was 33 months (95% CI: 27.7–38.3), with 43 months (95% CI: 33.1–52.9) for TACE and 26 months (95% CI: 18.0–34.0) for TARE. TACE patients had a longer mOS (37.0 months, (95% CI: 29.1–44.9) compared to 22.0 months (95% CI: 14.3–29.7) for TARE (HR 1.65; 95% CI: 1.19–2.29, *p* = 0.002; Figure 1). Tumor response rates by mRECIST were similar (Table 4). In BCLC 0/A patients, complete response rates were higher with TACE (43.2%) vs. TARE (34.3%, *p* = 0.012), but tumor response rates were similar in BCLC B patients.

Overall, BCLC 0/A patients had longer mOS (41.0 months; 95% CI: 13.5–68.5) than BCLC B patients (25.0 months; 95% CI: 18.7–31.3, *p* = 0.019) (Appendix A). Child–Pugh A patients had a longer mOS (34 months; 95% CI: 27.5–40.5) than Child–Pugh B patients (15 months; 95% CI: 8.0–22.0, *p* < 0.001) (Appendix A).

Over a median follow-up of 47 months, BCLC 0/A patients treated with TACE had a longer mOS of 60 months (95% CI: 42.1–77.9) vs. 25 months (95% CI: 17.0–33.0) for TARE (HR 2.35; 95% CI: 1.17–4.69, *p* = 0.016) (Figure 2A). Similarly, over a median follow-up of 30 months, BCLC B patients treated with TACE showed a non-significant trend toward a longer mOS of 32 months (95% CI: 20.5–43.5) vs. 20 months (95% CI: 12.4–27.6) for TARE (HR 1.39; 95% CI: 0.96–2.03, *p* = 0.080) (Figure 2B).

Over a median of 38 months of follow-up, Child–Pugh A patients treated with TACE had a longer mOS of 44.0 months (95% CI: 37.2–50.8) vs. 24.0 months (95% CI: 18.5–29.5) for TARE (HR 1.65; 95% CI: 1.12–2.42, *p* = 0.003) (Appendix A). Similarly, over a median follow-up of 19 months, Child–Pugh B patients treated with TACE showed a non-significant trend toward a longer mOS with 21.0 months (95% CI: 14.7–27.3) vs. 8.0 months (95% CI: 4.6–11.5) for TARE (HR 1.82; 95% CI: 0.95–3.45, *p* = 0.059) (Appendix A).

### 3.3. Safety

Median interventions were similar in both treatment cohorts (TACE: 1, range 1–3; TARE: 1, range 1–2). In the TACE group, 24 patients (13.9%) received ≥2 interventions, while 10 patients (9.6%) in the TARE group received 2 interventions. Overall, TRAEs within one week were significantly higher with TACE (64.0%) vs. TARE (33.7%, *p* < 0.001). The most common adverse events were gastrointestinal symptoms, fever, and elevated aminotransferases (Appendix A). Grade 3/4 TRAEs were comparable, including TACE-related cholecystitis (n = 2), TARE-related gastric ulceration (n = 1), and acute liver failure (n = 1 each). No procedure-related deaths occurred. Hepatic decompensation within 6 months was more frequent with TARE- (26.0%) vs. TACE-treated (13.7%, *p* = 0.010), driven by new-onset ascites (Appendix A).

## 4. Discussion

This multicenter study compared TACE and TARE in uHCC, demonstrating TACE’s superiority. Baseline characteristics were balanced, although a slightly higher proportion of patients with BCLC stage B disease received TARE (69.2% vs. 56.6%), reflecting routine clinical practice at the participating tertiary care centers, where TARE is often used to control and downgrade disease while awaiting liver transplant surgery when feasible. TACE yielded a longer mOS (37 vs. 22 months, HR 1.65; 95% CI: 1.19–2.29, *p* = 0.002). Complete response rates per mRECIST in different BCLC stages were higher for TACE than TARE, especially in BCLC 0/A (43.2% vs. 34.3%, *p* = 0.012). These outcomes are critical for uHCC, where treatment goals include tumor control and prolonging survival [3].

The TARE-related survival observed in this analysis (BCLC 0/A: 25 months; BCLC B: 20 months) aligns with a multicenter European study (24.4 and 16.9 months for early- and intermediate-stage HCC, respectively) [6]. Similarly, the mOS of patients undergoing TACE in BCLC stage 0/A and B disease was similar to that reported by Wang et al. [4]. However, two randomized phase 2 trials showed mixed outcomes [7,8]. Dhondt et al. reported a longer mOS with TARE (30.2 vs. 15.6 months for DEB-TACE) [7]. Conversely, Salem et al. found a similar mOS for cTACE (17.7 months) and TARE (18.6 months; *p* = 0.99) [8]. Nonetheless, these trials’ conclusions are limited by single-center designs and unexpectedly low TACE survival [9,10].

Real-world comparative data are scarce. Our findings contrast with previous studies and meta-analyses, showing that TARE-related survival is superior to TACE [11,12,13]. For example, Kim et al. reported longer TARE survival (HR 0.54; 95% CI: 0.31–0.92; *p* = 0.02) [12]. A recent meta-analysis also suggested improved survival with TARE [13]. Conversely, comparable survival has been reported between TACE and TARE [14,15,16]. Notably, a meta-analysis showed similar effects in terms of survival and response rate between the two modalities, although tumor progression was delayed after radioembolization [17]. These studies often included heterogeneous BCLC A–D stages and Child–Pugh A–C cohorts. Our study, the largest comparative cohort, uniquely evaluates responses by BCLC stage and Child–Pugh status. This granular analysis confirms TACE’s survival advantage, particularly in BCLC 0/A and Child–Pugh A patients. These findings, however, must be viewed in the context of the greater tumor burden in our TARE cohort and its potential impact on disease control and survival. While previous studies have shown that tumor burden predicts outcomes in patients undergoing TACE [4,9], conversely, in TARE, larger tumor number or size has not been consistently predictive of survival [18,19,20,21]. In the multicenter DOSISPHERE-01 trial utilizing personalized dosimetry with Y90, the objective response rate to the index lesion was 71%, despite exhibiting a mean tumor diameter of 10 cm [19]. Likewise, in the LEGACY study, Salem et al. evaluated TARE for solitary HCC lesions ≤8 cm. They observed an objective response of 88.3% and a 3-year OS of 86.6% [21]. Taken together, these findings suggest that tumor diameter may play a greater prognostic role in TACE-treated patients than in those receiving TARE. However, this hypothesis warrants further investigation, ideally through a comparative analysis of these two LRTs in patients with smaller tumors (diameter < 4 cm), where the influence of size on treatment response can be more accurately assessed.

Similarly, BCLC B patients showed a 12-month survival difference in favor of TACE, which, although not statistically significant, may be clinically meaningful. An mOS of 30 months in TACE-treated patients aligns with the target outcomes proposed in the BCLC treatment algorithm (>30 months) [3]. However, few studies with TACE in BCLC stage B disease reach target survival outcomes, largely as a consequence of including patients with large tumor burden, limited hepatic reserve, or poor performance status [4,5]. Therefore, the identification of negative prognostic features in routine clinical practice is paramount for maximizing the treatment benefits of LRTs in BCLC stage B patients.

While Child–Pugh A patients had a longer survival (34 months), Child–Pugh B patients had a shorter survival of 15 months, consistent with previous reports [5,16,22]. TACE outperformed TARE across these groups, although the survival benefit was non-significant in Child–Pugh B. Such differences in survival related to the Child–Pugh class could potentially be in part due to the competing risk of mortality from progressive liver dysfunction in decompensated patients, not allowing for sufficient time to avail of the benefits of LRT. Indeed, Allaire et al. showed that the occurrence of hepatic decompensation after TACE precluded further access to HCC treatment in 78% of patients [23].

Previous studies have suggested that outcomes could improve with super-selective techniques, radiation segmentectomy, DEB-TACE, and operator expertise [24,25]. These factors are unlikely to explain our survival differences. Centers participating in our study provide tertiary care, with vast experience in the management of HCC, including access to LT, and with greater than 15 years’ experience with both LRT procedures. Therefore, operator experience or technique is unlikely to account for these differences in outcome. More importantly, a recent randomized phase 2 study has demonstrated that personalized dosimetry in TARE significantly improved outcomes compared to standard dosimetry [19]. Such refinements in Y90 embolization were not incorporated until recently in our cohort and could have impacted treatment outcomes. Crucially, in an evolving field of Y90 treatment, the value of such modifications in tumor absorbed dose must be further clarified in larger randomized clinical trials.

TARE patients had higher AFP (>400 ng/mL) and BCLC B disease, both of which are known survival predictors [26]. Cohorts were otherwise matched for age, comorbidities, Child–Pugh class, and performance status. In previous analyses, Child–Pugh score, BCLC stage, and performance status were prognostically valuable to predict OS in uHCC [4,5,20]. Survival varied by Child–Pugh (17.2 vs. 7.7 months for A vs. B) and BCLC stage (26.9 vs. 17.2 months for A vs. B) [5]. TACE-related survival in BCLC B and Child–Pugh B (32–34 months) aligns with prior real-world data [4]. On the other hand, the role of higher AFP (>400 ng/mL) levels impacting treatment outcomes in our analysis warrants further exploration. In retrospective analyses, it would be difficult to ascertain the rationale for assigning patients with somewhat higher AFP levels to the TARE intervention. The reasons for this could include a perceived aggressive biology of the tumor leading to physician bias in favor of TARE intervention or TARE enrollment in non-transplant centers, which are more prone to managing patients with HCC having higher AFP levels. Since robust cut-offs for higher AFP levels are generally not available to ascertain the suitability of one form of therapy over another, an individualized patient profile may determine treatment decisions, including whether TACE, TARE, or systemic therapy should be favored in such situations [3]. Nonetheless, such confounding effects can be more adequately addressed in large randomized controlled clinical trials where patient stratification can be performed by AFP levels and other key prognostic indicators.

Disease control rates were comparable in both LRT cohorts. TACE achieved higher complete response rates in BCLC 0/A (43.2% vs. 34.3%, *p* = 0.012). Similarly to this analysis, several studies have reported comparable disease control rates in either LRT modality [7,8,14]. A meta-analysis by Facciorusso et al. also supported these findings, concluding that TARE and TACE showed similar response rates despite slower time to tumor progression with TARE [17]. On the other hand, a recent meta-analysis suggested higher disease control with TARE [13]. However, liver decompensation in our study was significantly higher with TARE-treated patients (26.0% vs. 13.7%, *p* = 0.010), driven by ascites, and in line with previous reports [27,28]. Previous studies have shown that hepatic reserve is critical for uHCC survival [29], and the development of post-TARE hepatic decompensation was shown to reduce mOS (8 vs. 38 months) [30]. TARE-related decompensation may stem from disease progression or radioembolization-induced liver disease (REILD), a condition characterized by hepatic sinusoidal obstruction, causing hyperbilirubinemia and/or ascites within 2 months [31]. In a systematic review, Braat et al. reported REILD incidence ranged from 0% to 31% [32]. The competing risk of mortality due to progressive liver decompensation, which in turn precludes further access to HCC treatment [23], may help explain the lower survival outcomes observed in TARE-treated patients in our study. Towards this, appropriate patient selection and identification of predictors of liver decompensation and mortality following TARE are essential for optimizing LRT management strategies. Furthermore, we included patients who had received TACE as a bridge to LT. An alternative approach would have been to exclude such patients or censor them at the time of transplant surgery. However, given the similar rates of LT performed post both LRTs (8.6% vs. 8.7%), this would likely have had minimal impact on overall survival outcomes.

This retrospective study has limitations that are inherent to retrospective analyses, including selection bias and confounding. Unlike clinical trials focusing on Child–Pugh A patients to avoid competing risks of death from cirrhosis on the overall outcome, our real-world data reflect heterogeneous patients, with diversity in patient selection from participating centers, providing valuable information for evaluating the effectiveness and safety of these two primary LRTs. On the other hand, the heterogeneity of BCLC B complicates interpretation due to technical and physician biases. Clearly, stratifying patients by BCLC stage would help mitigate biases in data interpretation. Larger, multicenter randomized controlled trials, with well-defined categorization of BCLC 0/A and B disease, are needed to validate these findings. However, inherent difficulties in controlling for confounders, such as tumor size, location, and vascularity, are difficult to account for and feasibly address in retrospective analyses, or even in prospective ones. Such subtle distinctions in LRT selection are subject to the expertise and preferences of treating physicians or interventional radiologists, making the process vulnerable to selection bias. Additionally, the absence of stratification and baseline differences in comorbidities, such as diabetes, tumor burden (size and number), AFP level, and BCLC stage, may further contribute to bias. It is possible that TARE patients may have been TACE-unsuitable. TARE’s newer adoption may lead to suboptimal patient selection. Highlighting this point, the LEGACY study has demonstrated TARE’s enhanced efficacy in early-stage HCC with small, single lesions, suggesting that patient selection is crucial in optimizing treatment outcomes [21].

## 5. Conclusions

These real-life data confirm TACE as the preferred therapy for unresectable HCC, offering significantly improved OS compared to TARE. However, these results must be interpreted in the context of the greater tumor burden and more advanced disease stage observed in patients treated with TARE. The findings advocate for a more nuanced selection of LRTs in patients with HCC who are unsuitable for curative interventions. TACE should be prioritized for early- and intermediate-stage HCC, and further randomized trials are needed to refine the optimal role of TARE, particularly in patients with higher tumor burden or those unsuitable for TACE.

## Figures and Tables

**Figure 1 cancers-17-02254-f001:**
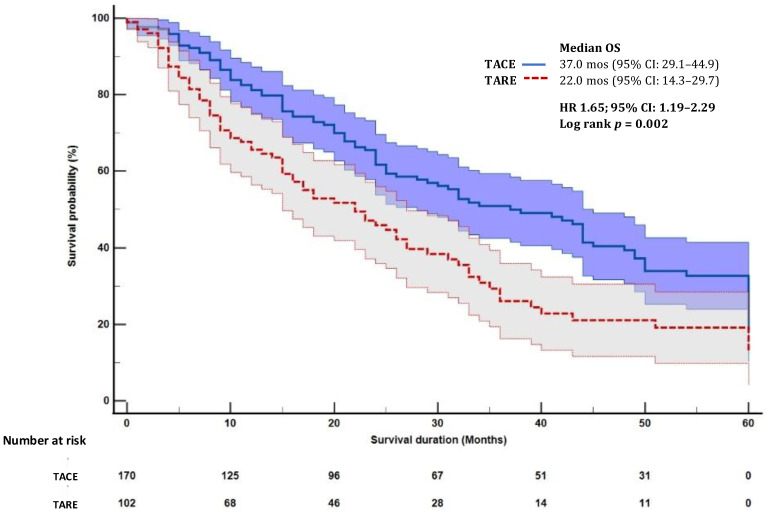
Survival analysis of patients undergoing transarterial chemoembolization and radioembolization.

**Figure 2 cancers-17-02254-f002:**
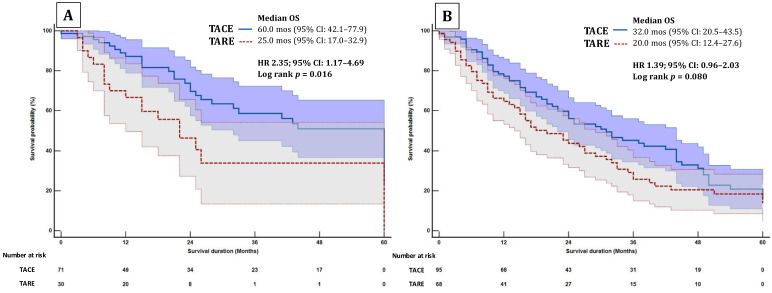
Survival analysis of patients undergoing transarterial chemoembolization and radioembolization in sub-groups of (**A**) BCLC 0/A and (**B**) BCLC B.

**Table 1 cancers-17-02254-t001:** Baseline characteristics of hepatocellular carcinoma patients in the TACE and TARE cohorts.

Parameter	Total n = 279	TACE n = 175	TARE n = 104	*p*
**Age** (yrs)	67.6 ± 10.1	67.3 ± 10.4	67.9 ± 9.4	0.519
**Male gender**	207 (74.2)	128 (73.1)	79 (76.0)	0.603
**BMI** (kg/m^2^)	27.5 ± 6.0	27.8 ± 6.3	27.0 ± 5.5	0.528
**Diabetes**	166 (59.5)	97 (55.4)	69 (66.3)	0.072
**Bilirubin** (μmol/L)	13.0 (8.6–19.7)	12.3 (8.8–20.4)	13.5 (8.3–19.0)	0.950
**ALT** (U/L)	30.0 (19.0–45.2)	30.0 (18.0–46.0)	31.5 (21.0–44.9)	0.368
**AST** (U/L)	36.0 (25.0–57.0)	35.0 (24.0–56.0)	39.0 (27.0–60.2)	0.147
**ALP** (U/L)	124.0 (90.8–175.0)	120.0 (89.8–164.5)	129.0 (92.5–195.8)	0.260
**GGT** (U/L)	108.0 (55.0–208.7)	99.0 (51.0–198.0)	125.8 (66.5–212.2)	0.222
**Albumin** (g/L)	36.0 (31.0–39.0)	36.0 (31.0–39.0)	36.0 (31.2–39.2)	0.808
**INR**	1.1 (1.1–1.2)	1.1 (1.1–1.2)	1.1 (1.1–1.2)	0.365
**Platelets** (10^9^/L)	160.5 (111.5–230.3)	161.5 (104.8–230.3)	159.0 (117.5–230.0)	0.621
**AFP** (ng/mL)	11.3 (4.4–59.5)	9.3 (4.0–45.7)	12.0 (5.0–160.0)	0.153
AFP < 400	236 (85.5)	154 (89.0)	82 (79.6)	0.032
AFP ≥ 400	40 (14.5)	19 (11.0)	21 (20.4)	
**Etiology**				0.158
Hepatitis B	88 (31.5)	55 (31.4)	33 (31.7)	
Hepatitis C	87 (31.2)	58 (33.1)	29 (27.9)	
MASLD	70 (25.1)	37 (21.1)	33 (31.7)	
Other	34 (12.2)	25 (14.3)	09 (8.7)	
**Non-Cirrhotic**	8 (2.9)	4 (2.3)	4 (3.8)	0.477
**Cirrhosis CTP A**	202 (72.4)	124 (70.9)	78 (75.0)	
**Cirrhosis CTP B**	69 (24.7)	47 (26.9)	22 (21.2)	
CTP score 5/6	202 (74.5)	124 (72.5)	78 (78.0)	0.668
CTP score 7	44 (16.2)	30 (17.5)	14 (14.0)	
CTP score 8/9	25 (9.2)	17 (9.9)	8 (8.0)	
**ECOG**				0.130
PS 0–1	244 (87.5)	149 (85.1)	95 (91.3)	
PS 2	35 (12.5)	26 (14.9)	09 (8.7)	
**BCLC Stage**				0.036
Stage 0/A	108 (38.7)	76 (43.4)	32 (30.8)	
Stage B	171 (61.3)	99 (56.6)	72 (69.2)	
**Tumors**				
Number	1.0 (1.0–2.0)	1.0 (1.0–1.0)	2.0 (1.0–3.0)	<0.001
Diameter, largest (cm)	3.7 (2.5–5.7)	3.1 (2.0–4.2)	4.4 (2.8–6.4)	<0.001

Data presented as mean ± SD, median (IQR), or n (%) as relevant; TACE, transarterial chemoembolization; TARE, transarterial radioembolization; BMI, body mass index, ALT, alanine aminotransferase; AST, aspartate aminotransferase; GGT, gamma glutamyl transpeptidase; ALP, alkaline phosphatase; MASLD, metabolic dysfunction-associated steatotic liver disease; CTP, Child–Turcotte–Pugh; ECOG, Eastern Cooperative Oncology Group; PS, performance status; BCLC, Barcelona Clinic Liver Cancer.

**Table 2 cancers-17-02254-t002:** Baseline characteristics of patients in the TACE and TARE cohorts in early-stage (BCLC 0/A) hepatocellular carcinoma.

Parameter	Total n = 108	TACE n = 76	TARE n = 32	*p*
**Age** (yrs)	67.8 ± 11.4	66.8 ± 11.5	70.3 ± 10.8	0.290
**Male gender**	73 (67.6)	51 (67.1)	22 (68.8)	0.716
**BMI** (kg/m^2^)	28.0 ± 6.5	27.9 ± 6.7	28.0 ± 6.2	0.926
**Diabetes**	73 (67.6)	50 (65.8)	23 (71.9)	0.537
**Bilirubin** (μmol/L)	12.0 (8.4–18.0)	12.0 (8.5–19.3)	12.0 (8.3–17.9)	0.856
**ALT** (U/L)	26.5 (17.3–41.0)	25.5 (17.0–41.0)	28.5 (18.3–42.3)	0.155
**AST** (U/L)	34.5 (22.3–55.8)	34.5 (22.0–55.8)	34.5 (23.3–56.3)	0.072
**ALP** (U/L)	118.5 (86.3–169.0)	118.5 (86.3–165.5)	120.0 (84.0–197.5)	0.278
**GGT** (U/L)	101.0 (54.0–218.0)	99.5 (49.3–203.0)	129.6 (63.8–230.5)	0.228
**Albumin** (g/L)	36.0 (31.0–40.0)	36.0 (31.0–40.0)	35.0 (29.8–38.8)	0.367
**INR**	1.1 (1.0–1.2)	1.1 (1.0–1.2)	1.1 (1.0–1.2)	0.471
**Platelets** (10^9^/L)	143.0 (101.0–209.0)	140.0 (99.0–215.0)	161.0 (106.5–205.5)	0.621
**AFP** (ng/mL)	8.0 (4.0–25.5)	8.0 (4.0–23.3)	8.9 (3.1–51.7)	0.046
AFP < 400	97 (91.5)	70 (93.3)	27 (87.1)	0.443
AFP ≥ 400	9 (8.5)	5 (6.7)	4 (12.9)	
**Etiology**				0.343
Hepatitis B	26 (24.1)	20 (26.3)	06 (18.8)	
Hepatitis C	41 (38.0)	31 (40.8)	10 (31.3)	
MASLD	31 (28.7)	18 (23.7)	13 (40.6)	
Other	10 (9.3)	7 (9.2)	3 (9.4)	
**Non-Cirrhotic**	2 (1.9)	0 (0)	2 (6.3)	0.133
**Cirrhosis CTP A**	83 (78.3)	60 (78.9)	23 (71.9)	
**Cirrhosis CTP B**	23 (21.7)	16 (21.1)	7 (21.9)	
CTP score 5/6	83 (78.3)	60 (78.9)	23 (76.7)	0.673
CTP score 7	16 (15.1)	12 (15.8)	4 (13.3)	
CTP score 8/9	7 (6.6)	4 (5.3)	3 (10.0)	
**ECOG**				0.102
PS 0–1	101 (93.5)	69 (90.8)	32 (100.0)	
PS 2	7 (6.5)	7 (9.2)	0 (0.0)	
**Tumors**				
Number	1.0 (1.0–1.0)	1.0 (1.0–1.0)	1.0 (1.0–1.0)	0.609
Diameter, largest (cm)	3.1 (2.0–4.2)	2.7 (1.8–3.7)	4.2 (2.3–5.8)	0.001

Data presented as mean ± SD, median (IQR), or n (%) as relevant; TACE, transarterial chemoembolization; TARE, transarterial radioembolization; BMI, body mass index, ALT, alanine aminotransferase; AST, aspartate aminotransferase; GGT, gamma glutamyl transpeptidase; ALP, alkaline phosphatase; MASLD, metabolic dysfunction-associated steatotic liver disease; CTP, Child–Turcotte–Pugh; ECOG, Eastern Cooperative Oncology Group; PS, performance status; BCLC, Barcelona Clinic Liver Cancer.

**Table 3 cancers-17-02254-t003:** Baseline characteristics of patients in the TACE and TARE cohorts in intermediate-stage (BCLC B) hepatocellular carcinoma.

Parameters	Total n = 171	TACE n = 99	TARE n = 72	*p*
**Age** (yrs)	67.4 ± 9.2	67.8 ± 9.6	66.9 ± 8.7	0.814
**Male gender**	134 (78.4)	77 (77.8)	57 (79.2)	0.828
**BMI** (kg/m^2^)	27.2 ± 5.7	27.6 ± 6.0	26.5 ± 5.2	0.381
**Diabetes**	93 (54.4)	47 (47.5)	46 (63.9)	0.033
**Bilirubin** (μmol/L)	13.5 (8.8–20.5)	13.0 (8.8–22.0)	13.7 (8.1–20.1)	0.956
**ALT** (U/L)	31.0 (21.0–48.0)	31.0 (20.0–48.0)	32.5 (22.3–48.1)	0.650
**AST** (U/L)	37.0 (26.4–57.0)	35.0 (25.0–56.6)	39.5 (28.0–63.0)	0.144
**ALP** (U/L)	127.6 (92.9–175.5)	120.5 (92.5–165.5)	131.0 (95.5–194.3)	0.346
**GGT** (U/L)	112.0 (55.5–188.0)	97.0 (51.5–208.0)	122.0 (66.0–173.0)	0.355
**Albumin** (g/L)	38.0 (31.7–39.0)	36.8 (31.7–39.0)	36.0 (31.2–39.6)	0.739
**INR**	1.1 (1.1–1.3)	1.2 (1.1–1.3)	1.1 (1.1–1.2)	0.313
**Platelets** (10^9^/L)	167.0 (117.0–245.0)	174.0 (108.0–246.0)	159.0 (120.0–244.8)	0.827
**AFP** (ng/mL)	13.2 (5.0–85.0)	12.9 (5.0–61.3)	14.2 (5.9–287.0)	0.258
AFP < 400	139 (81.8)	84 (85.7)	55 (76.4)	0.120
AFP ≥ 400	31 (18.2)	14 (14.3)	17 (23.6)	
**Etiology**				0.232
Hepatitis B	62 (36.3)	35 (35.4)	27 (37.5)	
Hepatitis C	46 (26.9)	27 (27.3)	19 (26.4)	
MASLD	39 (22.8)	19 (19.2)	20 (27.8)	
Other	24 (14.0)	18 (18.2)	6 (8.3)	
**Non-Cirrhotic**	6 (3.5)	4 (4.0)	2 (2.8)	0.235
**Cirrhosis CTP A**	119 (69.6)	64 (64.6)	55 (76.4)	
**Cirrhosis CTP B**	46 (26.9)	31 (31.3)	15 (20.8)	
CTP score 5/6	120 (72.7)	65 (68.4)	55 (78.6)	0.290
CTP score 7	27 (16.4)	17 (17.9)	10 (14.3)	
CTP score 8/9	18 (10.9)	13 (13.7)	5 (7.1)	
**ECOG**				0.243
PS 0–1	143 (83.6)	80 (80.8)	63 (87.5)	
PS 2	28 (16.4)	19 (19.2)	9 (12.5)	
**Tumors**				
Number	2.0 (1.0–3.0)	2.0 (1.0–3.0)	2.0 (1.0–3.0)	0.440
Diameter, largest (cm)	4.4 (2.9–6.4)	4.0 (3.0–5.6)	5.0 (2.7–7.8)	0.049

Data presented as mean ± SD, median (IQR), or n (%) as relevant; TACE, transarterial chemoembolization; TARE, transarterial radioembolization; BMI, body mass index, ALT, alanine aminotransferase; AST, aspartate aminotransferase; GGT, gamma glutamyl transpeptidase; ALP, alkaline phosphatase; MASLD, metabolic dysfunction-associated steatotic liver disease; CTP, Child–Turcotte–Pugh; ECOG, Eastern Cooperative Oncology Group; PS, performance status; BCLC, Barcelona Clinic Liver Cancer.

**Table 4 cancers-17-02254-t004:** Tumor response in the overall study population, and sub-cohorts of early- and intermediate-stage hepatocellular carcinoma according to modified Response Evaluation Criteria in Solid Tumors (mRECIST).

Tumor Response	Overall Cohort		*p*
	TACE (n = 175)	TARE (n = 104)	
Complete response	58 (33.9)	25 (24.3)	0.341
Partial response	55 (32.2)	42 (40.8)	
Stable disease	27 (15.8)	17 (16.5)	
Progressive disease	31 (18.1)	19 (18.4)	
**Tumor Response**	**Early-stage (BCLC 0/A) Cohort**		* **p** *
	**TACE (n = 76)**	**TARE (n = 32)**	
Complete response	32 (43.2)	11 (34.3)	0.012
Partial response	14 (18.9)	16 (50.0)	
Stable disease	15 (20.3)	3 (9.4)	
Progressive disease	13 (17.6)	2 (6.3)	
**Tumor Response**	**Intermediate-stage (BCLC B) Cohort**		* **p** *
	**TACE (n = 99)**	**TARE (n = 72)**	
Complete response	26 (26.8)	14 (19.7)	0.363
Partial response	41 (42.3)	26 (36.6)	
Stable disease	12 (12.4)	14 (19.7)	
Progressive disease	18 (18.6)	17 (23.9)	

Data presented as n (%); TACE, transarterial chemoembolization; TARE, transarterial radioembolization; BCLC, Barcelona Clinic Liver Cancer.

## Data Availability

The data that support the findings of this study are available on request from the corresponding author [F.M.S.]. The data are not publicly available due to [restrictions arising from containing information that could compromise the privacy of research participants]. Release of raw data would require express approval from institutional review boards governing the conduct of this study.

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
