# Peer review of "Transarterial Chemoembolization Outperforms Radioembolization in Early- and Intermediate-Stage Hepatocellular Carcinoma: A Multicenter Retrospective Study"

_cancers, 2025, doi:10.3390/cancers17132254_

Round 1
Reviewer 1 Report
Comments and Suggestions for Authors
Major revision recommendation
The study presents valuable insights into the comparative efficacy of TACE and TARE for early- and intermediate-stage HCC. However, there are several areas for improvement. The retrospective nature of the study introduces inherent biases, and there is a lack of stratification for critical variables such as AFP levels and coexisting comorbidities, which could confound the results. Additionally, a more thorough explanation of the patient selection criteria and the clinical implications of hepatic decompensation is needed. Future prospective studies with a more homogeneous cohort would enhance the validity of these findings.
- Why was the BCLC stage stratification not performed at the individual patient level, considering the significant impact of this classification on prognosis?
- How do you explain the difference in AFP levels between the two groups in the BCLC 0/A cohort? Could this have confounded the results?
- Was the decision to treat patients with TACE or TARE influenced by tumor location, size, or vascularity? If so, could this introduce selection bias?
- Were there any additional adjustments made for ECOG performance status, and how might this have influenced the interpretation of survival outcomes?
- In the survival analysis, did you account for possible competing risks, such as liver decompensation or progression to Child-Pugh C status?
- The study found significant differences in mOS. Could you elaborate on the clinical relevance of a 15-month difference in survival in the BCLC B group?
- What was the role of radiological imaging in post-treatment assessment, and how consistent were the imaging protocols across centers?
- Were there any biases in how tumor response was evaluated using mRECIST, and how might this affect the comparison of TACE and TARE?
- The study mentions that TARE patients were more likely to have BCLC B. How could this impact the generalizability of the results to a broader HCC population?
- Did the study control for the number of prior treatments, if any, patients had undergone? How might prior therapy have influenced outcomes?
Major revision recommendation
The study presents valuable insights into the comparative efficacy of TACE and TARE for early- and intermediate-stage HCC. However, there are several areas for improvement. The retrospective nature of the study introduces inherent biases, and there is a lack of stratification for critical variables such as AFP levels and coexisting comorbidities, which could confound the results. Additionally, a more thorough explanation of the patient selection criteria and the clinical implications of hepatic decompensation is needed. Future prospective studies with a more homogeneous cohort would enhance the validity of these findings.
- Why was the BCLC stage stratification not performed at the individual patient level, considering the significant impact of this classification on prognosis?
- How do you explain the difference in AFP levels between the two groups in the BCLC 0/A cohort? Could this have confounded the results?
- Was the decision to treat patients with TACE or TARE influenced by tumor location, size, or vascularity? If so, could this introduce selection bias?
- Were there any additional adjustments made for ECOG performance status, and how might this have influenced the interpretation of survival outcomes?
- In the survival analysis, did you account for possible competing risks, such as liver decompensation or progression to Child-Pugh C status?
- The study found significant differences in mOS. Could you elaborate on the clinical relevance of a 15-month difference in survival in the BCLC B group?
- What was the role of radiological imaging in post-treatment assessment, and how consistent were the imaging protocols across centers?
- Were there any biases in how tumor response was evaluated using mRECIST, and how might this affect the comparison of TACE and TARE?
- The study mentions that TARE patients were more likely to have BCLC B. How could this impact the generalizability of the results to a broader HCC population?
- Did the study control for the number of prior treatments, if any, patients had undergone? How might prior therapy have influenced outcomes?
Author Response
Reviewer 1:
Major revision recommendation
The study presents valuable insights into the comparative efficacy of TACE and TARE for early- and intermediate-stage HCC. However, there are several areas for improvement. The retrospective nature of the study introduces inherent biases, and there is a lack of stratification for critical variables such as AFP levels and coexisting comorbidities, which could confound the results. Additionally, a more thorough explanation of the patient selection criteria and the clinical implications of hepatic decompensation is needed. Future prospective studies with a more homogeneous cohort would enhance the validity of these findings.
Comment 1: Why was the BCLC stage stratification not performed at the individual patient level, considering the significant impact of this classification on prognosis?
Response: We thank the reviewer for this comment. We categorized each patient’s BCLC stage (0/A and B) at study entry and analyzed treatment response outcomes separately to account for differences in treatment disposition by physician/MDT preference. Due to the retrospective design, blinded randomization based on BCLC stage was not feasible. We have expanded the Discussion to highlight this limitation, noting challenges in controlling for confounders like tumor size and vascularity (page 19).
Comment 2. How do you explain the difference in AFP levels between the two groups in the BCLC 0/A cohort? Could this have confounded the results?
Response: We thank the reviewer for this observation. Our results show that baseline median AFP levels and the proportion of patients with AFP >400 ng/mL were higher in the TARE cohort. It is largely a reflection of the MDT preference for TARE in patients with perceived higher-risk features, though the retrospective design limits definitive conclusions. We have added a paragraph in the Discussion to emphasize AFP’s association with survival outcomes (page 17).
Comment 3. Was the decision to treat patients with TACE or TARE influenced by tumor location, size, or vascularity? If so, could this introduce selection bias?
Response: Treatment allocation was determined by interventional radiologists and MDTs based on clinical factors, including tumor characteristics. We agree with the reviewer that this may introduce selection bias. We have clarified MDT roles in the Methodology (page 8) and addressed these factors and the limitations (page 19).
Comment 4. Were there any additional adjustments made for ECOG performance status, and how might this have influenced the interpretation of survival outcomes?
Response: We thank the reviewer for this comment. Baseline ECOG status was similar between cohorts (Table 1). Although the TACE group had slightly more ECOG 2 patients, the difference was not significant, and the small number of ECOG 2 patients did not warrant further stratification. We have clarified this in the Results (page 12).
Comment 5. In the survival analysis, did you account for possible competing risks, such as liver decompensation or progression to Child-Pugh C status?
Response: We have described in the results section the rate of hepatic decompensation and also discussed this in the relevant section. We have reported that liver decompensation occurred more frequently in TARE-treated patients, consistent with previous reports. Given the low incidence of decompensation, a competing risk analysis was unlikely to provide meaningful insights. We have expanded the Discussion to clarify these findings (page 18).
Comment 6. The study found significant differences in mOS. Could you elaborate on the clinical relevance of a 15-month difference in survival in the BCLC B group?
Response: We thank the reviewer for this important point which we failed to elaborate upon within the manuscript. We have now added in the Discussion section the points that address the clinical significance of this difference, and emphasized its potential to guide treatment decisions (page 15).
Comment 7. What was the role of radiological imaging in post-treatment assessment, and how consistent were the imaging protocols across centers?
Response: Follow up imaging was not consistent across the treatment centers for TACE while being generally consistent for TARE (12, 24 weeks as recommended). Physicians/centers performed post-treatment radiological imaging for TACE-treated patients anywhere between 6 and 12 weeks. As such we provided intervals (8-12 weeks for TACE and 12-24 weeks for TARE) for reporting treatment responses as mentioned within the methodology section. Subsequent imaging after the initial assessment was generally consistent with 3-monthly assessments. This has been clarified further in the methodology section (page 9).
Comment 8. Were there any biases in how tumor response was evaluated using mRECIST, and how might this affect the comparison of TACE and TARE?
Response: All imaging was reviewed by study radiologists blinded to clinical data and survival outcomes, using mRECIST criteria (page 9). While radiologists could not be blinded to intervention type, standardized mRECIST application minimized bias. This is clarified in the Methodology section.
Comment 9. The study mentions that TARE patients were more likely to have BCLC B. How could this impact the generalizability of the results to a broader HCC population?
Response: This was an important difference for patient treatment disposition seen at baseline. As such, we have separately analyzed and reported the outcomes of sub-groups BCLC 0/A and B to overcome the bias of generalizability to the broader HCC population. This would allow the reader to assess the impact of different interventions within the overall groups, and also to be able to scrutinize in greater detail as per the comprehensive breakdown of baseline data within the sub-groups as provided within tables 2 and 3, and the outcomes within table 4 and figure 2.
Comment 10. Did the study control for the number of prior treatments, if any, patients had undergone? How might prior therapy have influenced outcomes?
Response: We thank the reviewer for this comment, which the readers will certainly find useful within this manuscript. We have now provided, in extensive detail within the results section the prior procedures the study patients had undergone. The results show that additional interventions such as ablative therapies and resections were infrequent and similar in both treatment groups (page 12).
Reviewer 2 Report
Comments and Suggestions for Authors
The topic is of interest but i think there is a selection bias due to the significant difference in terms of tumoral stage and baseline AFP levels between the two groups. This may explain this unexpected result, that is the improve survival with TACE over TARE.
Please add the number at risk to the KM curves
The authors should comment more on the state of the art in this field (in this regard cite the SRMA: PMID: 27366304)
Author Response
The topic is of interest but i think there is a selection bias due to the significant difference in terms of tumoral stage and baseline AFP levels between the two groups. This may explain this unexpected result, that is the improve survival with TACE over TARE.
Response: We thank the Reviewer for the insightful feedback. We acknowledge concerns about selection bias due to differences in tumoral stage and AFP levels, which we have addressed through subgroup analyses, additional tumor characteristic data, and an expanded Discussion section (pages 15, 19). Below, we respond to each comment.
Comment 1. Please add the number at risk to the KM curves
Response 1: We regret this oversight and have added the number at risk to Figures 1 and 2.
Comment 2. The authors should comment more on the state of the art in this field (in this regard cite the SRMA: PMID: 27366304)
Response 2: We thank the reviewer for pointing out this important study; we have now included its findings within our Discussion section in two separate places, and cited it appropriately (pages 14, 17, 23).
Comment 3: There is a selection bias due to the significant difference in terms of tumoral stage and baseline AFP levels between the two groups.
Response 3: We acknowledge differences in tumoral stage and AFP levels between groups. To mitigate this, we analyzed BCLC 0/A and B subgroups separately and added tumor size and number data, showing larger tumors in the TARE group. These additions in the Discussion section enhance transparency and address potential confounding factors.
Reviewer 3 Report
Comments and Suggestions for Authors
Authors aimed to compare clinical efficacy between TACE vs. TARE for early- and intermediate-stage HCC.
This is an interesting paper.
There are several concerns to be addressed.
1) Please, provide the multivariable analysis.
2) Tumor number and size should be described in the comparison of baseline characteristics.
3) In the intermediate stage, some patients might have huge HCC, whereas othes might have multile but small HCC.
Is there any difference between TACE vs TARE ?
4) What about the next treatment modality after TACE or TARE ?
Author Response
Authors aimed to compare clinical efficacy between TACE vs. TARE for early- and intermediate-stage HCC. This is an interesting paper. There are several concerns to be addressed.
Response: We thank the Reviewer for the valuable comments. We have addressed concerns by adding tumor characteristics, subsequent treatment data, and clarifying the study’s scope, with updates reflected in the Results, Discussion, and Tables (pages 11, 15, 18, 19). Below, we respond to each point.
Comment 1. Please, provide the multivariable analysis.
Response 1: This is an interesting point raised by this reviewer. While multivariable analysis may help to scrutinize the independent factors responsible for treatment outcomes, however, the primary aim of this study was to compare the safety and efficacy of TACE and TARE in early- and intermediate-stage HCC. Adding tumor characteristics (size, number) to the Results and Tables 1–3 addresses confounding factors without necessitating multivariable analysis, which is beyond the scope of this manuscript (pages 11, 15).
Comment 2. Tumor number and size should be described in the comparison of baseline characteristics.
Response 2: We thank the reviewer for this insightful comment. We have now added these details, describing the size and number within the results section as well as in the Tables. These details were provided for the overall groups as well as in BCLC 0/A and BCLC B groups. In brief, maximum tumor diameter was larger in TARE group than in TACE (4.4 vs. 3.1 cm, p<0.001); these differences also persisted in the sub-groups of BCLC 0/A (2.7 vs. 4.2 cm, p=0.001) and BCLC B 4.0 vs. 5.0 cm, p=0.049). In light of these findings, we have modified the discussion section to explain the relevance of these findings and also within the study limitations section. We have also added this point within the conclusion section. Finally, we have added this finding within the results section of the abstract (pages 11, 15, 18 and 19; tables 1-3).
Comment 3. In the intermediate stage, some patients might have huge HCC, whereas others might have multiple but small HCC. Is there any difference between TACE vs TARE ?
Response 3: As mentioned in the above point, we have added these results also for BCLC B, including the size and number of lesions within the results section and tables, and explained the relevance of these findings.
Comment 4. What about the next treatment modality after TACE or TARE ?
Response 4: This is an important point raised by the reviewer. We have now added in extensive detail within the results section the subsequent procedures the patients underwent. In brief, subsequent numbers and type of interventions, including TACE and TARE, were similar in both treatment groups. In addition, we have also provided details on the number of patients who underwent subsequent liver transplantation in both groups, which were also similar. We have also discussed this within the Discussion section (pages 11, 18).
Round 2
Reviewer 1 Report
Comments and Suggestions for Authors
Accept in present form
Reviewer 2 Report
Comments and Suggestions for Authors
The manuscript is OK now. Thank you!
Reviewer 3 Report
Comments and Suggestions for Authors
Authors addressed raised issues appropriately.